# Immunogenicity of ChAdOx1 nCoV-19 Booster Vaccination Following Two CoronaVac Shots in Healthcare Workers

**DOI:** 10.3390/vaccines10020217

**Published:** 2022-01-30

**Authors:** Wisit Prasithsirikul, Krit Pongpirul, Tanawin Nopsopon, Phanupong Phutrakool, Wannarat Pongpirul, Chatpol Samuthpongtorn, Pawita Suwanwattana, Anan Jongkaewwattana

**Affiliations:** 1Bamrasnaradura Infectious Diseases Institute, Nonthaburi 11000, Thailand; wisit.p@bidi.mail.go.th (W.P.); awannarat@yahoo.com (W.P.); jamie15006@docchula.com (C.S.); pawitasuwan@bidi.mail.go.th (P.S.); 2Department of Preventive and Social Medicine, Faculty of Medicine, Chulalongkorn University, Bangkok 10330, Thailand; tnopsopon@hsph.harvard.edu; 3School of Global Health, Faculty of Medicine, Chulalongkorn University, Bangkok 10330, Thailand; 4Department of International Health, Johns Hopkins Bloomberg School of Public Health, Baltimore, MD 21205, USA; 5Clinical Research Center, Bumrungrad International Hospital, Bangkok 10110, Thailand; 6Harvard T.H. Chan School of Public Health, Harvard University, Boston, MA 02115, USA; 7Research Affairs, Faculty of Medicine, Chulalongkorn University, Bangkok 10330, Thailand; phanupong.p@chula.ac.th; 8National Center of Genetic Engineering and Biotechnology, Khlong Luang, Pathum Thani 12120, Thailand; anan.jon@biotec.or.th

**Keywords:** COVID-19, SARS-CoV-2, vaccines, immunogenicity, ChAdOx1, CoronaVac, third dose, booster, healthcare worker, cellular immune response

## Abstract

During the early phase of the COVID-19 pandemic, several countries, including Thailand, provided two shots of CoronaVac to healthcare workers. Whereas ChAdOx1 nCoV-19 is the promising vaccine as the booster dose, the data on immunogenicity when administered after CoronaVac have been limited. The purpose of this study was to evaluate the immunogenicity of ChAdOx1 nCoV-19 as the third dose vaccine in healthcare workers who previously received two shots of CoronaVac. The blood samples were obtained before the third vaccination dose, and one month and three months after vaccination. All participants were measured for humoral immunity including anti-spike IgG and neutralizing antibody by ELISA. Twenty participants were stratified by random samples based on baseline IgG status for a cellular immunity function test at three-month post-vaccination, which included T cell and B cell functions by ELISpot. This study showed significant improvement for both humoral and cellular immunity one month after vaccination. Subgroup analysis indicated a significantly higher neutralizing antibody improvement for the population with a negative anti-spike IgG at baseline. Our study suggests that, while immunity level declines at three months post-vaccination, the level was sufficiently high to protect against SARS-CoV-2.

## 1. Introduction

Front-line healthcare workers are a high-risk population for developing COVID-19, as there is substantial occupational exposure to SARS-CoV-2 from both patients [1] and co-workers [2], precipitated by inadequate personal protection in limited-resource settings [3]. A Significantly increased risk of testing positive for COVID-19 has been reported in front-line healthcare workers compared to the general population [4]. The prevalence of SARS-CoV-2 infection in healthcare workers was also higher than in non-healthcare workers during the early phase of the COVID-19 pandemic in the United States [5]. During the early pandemic in Thailand, a considerable number of healthcare workers tested positive for antibodies against COVID-19 in a province with no confirmed COVID-19 cases [6] and in community hospitals nationwide [7].

During the early phase of the COVID-19 pandemic, several countries, including Thailand, had limited access to mRNA or viral-vector vaccines. As an immediate step to contain a looming COVID-19 surge, the administration of CoronaVac, an inactivated vaccine developed by Sinovac Biotech, Beijing, China, for the general population and healthcare workers was implemented. While the high antibody responses after two shots of CoronaVac in adults have been documented [8,9], the antibodies against SARS-CoV-2 virus after CoronaVac immunization were substantially reduced within 3–6 months [10,11]. Moreover, the real-world data showed imperfect seroconversion, with approximately one-fourth of patients seronegative after receiving two shots of CoronaVac by 3–8 weeks, indicating the need for a booster shot for those who were fully vaccinated [12,13]. Recently, a randomized controlled trial on the homologous third dose of CoronaVac demonstrated significantly boosted immune responses against SARS-CoV-2 [14], but the immunogenicity was lower compared to the heterologous third dose of BNT162b2 after two shots of CoronaVac [15,16]. In addition, issues regarding the decline of the neutralizing activity against variants of concern induced by CoronaVac have been raised in several studies [17,18].

ChAdOx1 nCoV-19 (Vaxzevria, Cambridge, AstraZeneca, UK) as a third dose vaccine in the general population had high immunogenicity for both humoral and cellular arms of immune responses when administered following two shots of ChAdOx1 nCoV-19 [19,20] and BNT162b2 [20]. While robust immune responses, both humoral and cell-mediated immunity, were noted in systemic lupus erythematosus (SLE) patients boosted with ChAdOx1 nCoV-19 following two shots of CoronaVac [21], the immune profiles in the general population were available only for the humoral immunity [22]. For healthcare workers, the evidence on immunogenicity of ChAdOx1 nCov-19 as the third dose following two shots of CoronaVac was limited, and only a case report of intradermal ChAdOx1 nCov-19 vaccination following two shots of CoronaVac was available [23]. We herein aim to provide comprehensive evidence on the immunogenicity of ChAdOx1 nCov-19 as the third dose in healthcare workers who previously received two doses of CoronaVac.

## 2. Materials and Methods

### 2.1. Study Design and Population

In this prospective observational cohort study, we recruited 170 healthcare workers who received two shots of CoronaVac from February to March 2021 and were eligible for ChAdOx1 nCoV-19 vaccination as the third dose according to national health policy on the day of third dose vaccination at Bamrasnaradura Infectious Diseases Institute, Nonthaburi, Thailand—the national referral institute for infectious diseases.

Demographic characteristics and history of CoronaVac vaccination were obtained on the recruitment day using a case record form. Blood samples were obtained from all participants for SARS-CoV-2 spike Immunoglobulin G (IgG) and neutralizing antibody at the following time points: before ChAdOx1 nCoV-19 vaccination (baseline), and 1 month and 3 months after the third dose vaccination. For cellular immunity outcomes, participants were separated into quartiles (0–41.29, 41.57–64.94, 65.53–107.17, 110.31–772.95) based on the baseline SARS-CoV-2 IgG level, and then a total of 20 participants, consisting of five participants in each quartile, were selected for the cellular immunity outcome measurement at 3 months post-vaccination by stratified random sampling. The primary outcome for immunogenicity of ChAdOx1 nCoV-19 was SARS-CoV-2 spike IgG antibody, while secondary outcomes were SARS-CoV-2 neutralizing antibody by surrogate virus neutralization test, and cellular immune responses for both SARS-CoV-2-specific T cells and memory B cells.

The Ethics Committee of Research related to COVID-19 Disease or Public Health Emergency, Department of Disease Control, Ministry of Public Health, Thailand (Reference No. 64064; IRB.No. FWA00013622; 8 October 2021) approved this study, and written informed consent was obtained from all participants before enrollment.

### 2.2. Serological Analysis for Humoral and Cellular Immune Responses

SARS-CoV-2 spike IgG antibodies were measured by using ELISA with Anti-SARS-CoV-2 QuantiVac ELISA (IgG) agent (EUROIMMUN Medizinische Labordiagnostika AG, Lübeck, Germany), which measured antibodies against the spike protein quantitatively as binding antibody units (BAU) per ml as recommended by the World Health Organization, in which the antibody level is equal to or more than 32 BAU per ml, was considered positive for immunity against SARS-CoV-2.

Neutralizing antibodies were assessed using ELISA with SARS-CoV-2 NeutraLISA agent (EUROIMMUN Medizinische Labordiagnostika AG, Lübeck, Germany). The results were reported as the percent inhibition in which 35% or more was considered positive for neutralizing antibody against SARS-CoV-2. NeutraLISA was validated with the plaque reduction neutralization test (PRNT) with 98.6% agreement.

Cellular immune responses were enumerated using ELISpot (Mabtech AB, Nacka Strand, Sweden). For T cell functions, each test contained four ELISpot wells coated with anti-human interferon-gamma (IFN-γ) antibodies 1-D1K to bind with IFN-γ secreted by T cells. In addition, each ELISpot well contained peripheral blood mononuclear cells (PBMC) and test peptides to detect the T cell functions including SARS-CoV-2 S-defined peptide pool (Spike protein), SARS-CoV2 NMO-defined peptide pool (Nucleoprotein, Membrane protein, and Open reading frame proteins), Anti-CD3 monoclonal antibody as a positive control, and AIM V Medium as a negative control. The results were reported as spot forming cell (SFC) per 1.0 × 10^6^ T cells. For assessing memory B cells, ELISpot was used to measure antibody-secreting cells (ASCs) via a sandwich assay using specific recombinant antigens with stimulation reagents (R848 and IL-2).

### 2.3. Statistical Analysis

Descriptive statistics were used to present demographic data, continuous data were presented as mean with 95% confidence interval, and categorical data were presented as counts and percentages. The paired t-test was employed to compare the temporal differences of each participant. Subgroup analysis was conducted by the baseline SARS-CoV-2 Spike antibody level with cut-off 32 BAU/mL. The strength of difference between two subgroups was assessed using an unpaired t-test. Graph plots were generated using GraphPad Prism version 9.0.0 for Windows (GraphPad Software, San Diego, CA, USA). Statistical analysis was performed using STATA version 15 (College Station, TX, USA). A two-tailed *p* < 0.05 was considered statistically significant.

## 3. Results

### 3.1. Characteristics of Participants

This prospective cohort study included 170 healthcare workers who received ChAdOx1 nCoV-19 as the third dose following two shots of CoronaVac. None of the participants had a history of COVID-19 infection. Of 170 participants, 138 (81.81%) were female, aged 43.51 ± 10.52 years old (median 45, IQR 35–52). The participants were categorized into age groups: 19.41% were 46–50 years old, and 4.71% were 18–25 years old. Only eight participants were active smokers. There were 72 (42.35%) participants who had comorbidities (25 dyslipidemia, 22 hypertension, 14 diabetes mellitus, 10 allergy, 7 obesity, 4 thyroid, 4 anemia, 2 cardiovascular, 1 fatty liver, 1 asthma, 1 COPD, 1 migraine, 1 hepatitis B, 1 gastritis, and 1 psoriasis). Two (1.18%) participants had a history of vaccine allergy (Table 1).

### 3.2. Humoral Immunogenicity

All participants were tested for SARS-CoV-2 spike-specific antibody responses before ChAdOx1 nCoV-19 vaccination (baseline), and one month and three months post-vaccination. The mean anti-spike IgG level at baseline was 84.88 BAU/mL (95% confidence interval (CI), 71.61–98.14). At one month after the third dose vaccination, the mean anti-spike IgG level substantially increased to 2499.89 BAU/mL (95% CI, 2221.32–2778.47), which was significantly higher than the baseline IgG level (*p* < 0.001). After three months post-vaccination, the mean anti-spike IgG level decreased to 822.37 BAU/mL (95% CI, 728.53–916.21), which was notably lower than the IgG level at one month post-vaccination but still higher than the baseline level (*p* < 0.001) (Figure 1a).

For the neutralizing antibody, the mean percent of inhibition of RBD-ACE2 binding was 20.61% (95% CI, 17.73–23.49) at the baseline and significantly improved to 98.30% (95% CI, 97.11–99.50) at one month post-vaccination (*p* < 0.001). The mean inhibition percent was slightly reduced to 93.87% (95% CI 92.25–95.84) at three months post-vaccination, but the inhibition level was still significantly higher than the baseline (*p* < 0.001) (Figure 1b).

### 3.3. Humoral Immunogenicity Based on the Baseline Anti-Spike IgG Antibody Level

Subgroup analysis based on anti-spike IgG level before the third dose vaccination was performed, with a cut-off of more than 32 BAU/mL considered positive. Of 170 participants, 145 were positive for anti-spike IgG at baseline, and 25 were negative. In the negative IgG at the baseline group, the mean baseline anti-spike IgG antibody level was 0 BAU/mL and significantly increased to 1976.66 BAU/mL (95% CI, 1476.65–2476.66) after 1 month post-vaccination (*p* < 0.001), while in positive IgG at the baseline group, the mean baseline IgG was 98.20 BAU/mL (95% CI, 84.92–111.47) and significantly improved to 2576.61 BAU/mL (95% CI, 2263.95–2889.27) after 1 month post-vaccination (*p* < 0.001). While the improvement of anti-spike IgG level after 1 month of the third dose vaccination was slightly higher in those with positive IgG at the baseline, the improvement was not significantly different between participants with positive and negative baseline anti-spike IgG (*p* = 0.089). The anti-spike IgG at 3 months post-vaccination was significantly decreased to 635.22 BAU/mL (95% CI, 463.97–806.48) and 868.00 BAU/mL (95% CI, 765.88–970.12) in negative and positive baseline IgG groups, respectively, when compared to IgG level at 1 month post-vaccination. There was no significant difference of anti-spike IgG reduction at 3 months compared to 1 month post-vaccination between participants with positive and negative baseline anti-spike IgG (*p* = 0.117) (Figure 2).

The neutralizing antibody level 1 month after the third dose vaccination was significantly higher compared to before vaccination for both positive and negative anti-spike IgG at baseline groups with the improvement of inhibition percent from 8.83% (95% CI, 3.69–13.98) at baseline to 95.10% (95% CI, 86.92–103.29) in negative IgG group and the improvement from 21.40% (95% CI, 18.43–24.37) at baseline to 99.01% (95% CI, 98.79–99.23) in positive IgG group (*p* < 0.001). The improvement in inhibition percent was significantly higher in the negative IgG group than in the positive IgG group (*p* = 0.032). For the neutralizing antibody at 3 months post-vaccination, there was a significant reduction in inhibition percent compared with 1 month post-vaccination in both negative and positive anti-spike IgG at baseline groups, with a reduction in inhibition percent of 90.48% (95% CI, 82.47–98.48) in the negative IgG group and a reduction of 94.41% (95% CI, 93.12–95.69) in the positive IgG group (*p* < 0.001). However, the reduction in inhibition percent at 3 months after the third dose vaccination was not significantly different between the two subgroups (*p* = 0.982) (Figure 3).

### 3.4. Cellular Immunogenicity

The development of T cell function against SARS-CoV-2 spike protein after ChAdOx1 nCoV-19 as the third dose in participants who received two doses of CoronaVac was substantially higher in participants with positive anti-spike IgG at baseline with 143.06 SFC per one million cells (95% CI, 81.61–204.51) compared to 58.67 SRC per one million cells (95% CI, −35.77–53.11) in participants with negative anti-spike IgG at baseline. A similar trend was detected for the development of T cell function against SARS-CoV-2 NMO protein after the third dose vaccination, which was slightly higher in positive IgG at baseline group with 115.06 SFC per one million cells (95% CI, 77.34–152.77) compared to the mean level in negative IgG at baseline group with 85.33 SFC per one million cells (95% CI, −131.53–302.20). Nevertheless, there was an opposite trend for the development of B cell function against SARS-CoV-2, in which there was considerably higher B cell function in participants with negative anti-spike IgG at baseline with 396.67 SFC per one million cells (95% CI, 334.15–459.18) compared to the function in participants with positive IgG at baseline with 172.88 SFC per one million cells (95% CI, 68.70–277.06) (Figure 4).

## 4. Discussion

This prospective observational cohort study measured the immunogenicity of ChAdOx1 nCoV-19 as the third dose vaccination in healthcare workers who had previously received two doses of CoronaVac. The humoral immune responses were significantly increased 1 month after vaccination and subsequently waned 3 months after boosting for both participants with positive and negative baseline anti-spike IgG. However, the neutralizing antibody level at 1-month post-vaccination was notably higher in participants with positive baseline IgG, while no differences between two subgroups for anti-spike IgG improvement were observed. Additionally, the reduction in humoral immune response between the two subgroups was marginal and considered not significant. For the cellular immunity level, a higher T cell response in participants with positive anti-spike IgG at baseline was noted, while a higher B cell response in participants with negative IgG at baseline was detected.

The participant characteristics and the mean anti-spike IgG of 84.88 BAU/mL measured four months following the two-dose vaccination of CoronaVac in our study were comparable to that of other studies from Thailand: 92.9 BAU/mL (*n* = 185, median age 30 years, IQR 25–37, female 83.2%, measured three months) [10], 115 BAU/mL (*n* = 88, mean age 45.8 ± 9.3 years, female 60.2%, measured two months) [24], and 94.8 BAU/mL (*n* = 180, median age 35 years, IQR 29–44, female 84.2%, measured four weeks) [25]. Hence, our study population was representative for further investigation on the adjunctive immunogenicity of the ChAdOx1 nCoV-19 as a third booster shot.

The humoral immune response in this study showed similar results to the cohort study in the general population who received ChAdOx1 nCoV-19 as the third dose after two shots of CoronaVac with a shorter interval between the second dose of CoronaVac and the third dose ChAdOx1 nCoV-19 [22]. Moreover, a recent case report of a healthcare worker vaccinated intradermally with ChAdOx1 nCoV-19 as the third dose 6 weeks after the full course of CoronaVac also reported a slightly higher humoral response at 2 and 3 weeks post-vaccination than the result of this study [23]. Additionally, a similar improvement to this study in the humoral response in the case of SLE patients who received ChAdOx1 nCoV-19 after two shots of CoronaVac was recently documented [21]. A similar benefit of a third dose booster shot was seen in a randomized controlled trial using a regimen of three homologous CoronaVac doses [14], while previous observational studies showed that heterologous booster by BNT162b2 had higher immunogenicity for humoral immunity than homologous booster by CoronaVac in healthcare workers who were fully vaccinated with CoronaVac [15] and in older adults [16]. Immunogenicity at one month post vaccination of heterologous ChAdOx1 nCoV-19 as the third dose booster for both humoral and cellular responses in this study was similar to that of the homologous ChAOx1 nCoV-19 booster in participants who were fully vaccinated with two doses of ChAOx1 nCoV-19 in the sub-study of the randomized controlled trial [19]. On the other hand, a randomized controlled trial reported a poor improvement in cellular immunity in three homologous doses of ChAdOx1 nCoV-19, while indicating good improvement in cellular immunity at one month post-vaccination in heterologous ChAdOx1 nCoV-19 as the third booster dose in participants who received two shots of BNT162b2, which had a similar result to the heterologous ChAdOx1 nCoV-19 booster after being fully vaccinated with CoronaVac in this study.

While it is not clear how the antibody responses between groups with different IgG baseline correlate with the cellular immune response, we observed in this study that the activation of T and B cells is distinctly regulated in each group. The subjects in the positive IgG baseline showed remarkably higher T cell responses with both S and NMO peptide pools than those with negative IgG. In contrast, we found higher B cell responses in those with negative IgG at baseline, which is quite surprising, as IgG level has been believed to correlate with B cell response. To clarify this point further, we might need to increase the sample size for more in-depth analysis, as well as confirm these findings with other assays such as flow cytometry analysis of specific activation markers of T and B cells.

The optimal interval for the third booster dose for individuals who were fully vaccinated with CoronaVacis is still not fully established. The study on healthcare workers 18–59 years old who received two doses of CoronaVac showed a substantial reduction in humoral immunity at three months after the second shot of CoronaVac [10], while another single-center study showed the need for a third booster dose at two months after the second dose of CoronaVac, as up to 22.9% seronegativity in healthcare workers was observed, despite a complete course of vaccination [13]. A larger surveillance study indicated the highest IgG seropositivity at about 77.4% at 3 weeks after the second shot of CoronaVac and then declined to an average of 64.5% at 3–16 weeks after the second shot, which might suggest that this is an appropriate time for a booster shot [12]. The interval between the second shot of CoronaVac and the third dose of ChAdOx1 nCoV-19 in this study was based on the national policy and vaccine allocation during the study period, in which the interval time was approximately three months apart and provided promising immunogenicity for both humoral and cellular immunity.

While there was a decline in humoral and cellular immunity at three months after the ChAdOx1 nCoV-19 booster as the third dose in this study, the level of immunity was still sufficiently high for the known strains (i.e., Wuhan). Nonetheless, further investigation into whether immunity is protective against other emerging strains (i.e., Delta or Omicron) is required. Given that a new variant such as Omicron can substantially evade pre-existing immune responses, we speculate that the level of antibody after ChAdOx1 nCoV19 booster might not be sufficient for neutralizing this variant. Another booster vaccination is likely required for the healthcare worker subjects of this study. However, it remains to be determined whether another ChAdOx1 nCoV-19 shot or other platforms such as mRNA vaccine or recombinant subunit should provide the optimal level of protection against Omicron infection or disease in these subjects. These data will be highly useful for future management of COVID-19 booster shots to prepare for future emerging strains.

This study has several strengths. This study was the first cohort study on a population that was fully vaccinated with CoronaVac and received ChAdOx1 nCoV-19 as the third booster dose, which had the dynamic information of humoral and cellular immunity. The population included in this study was a high-risk population, which should be prioritized for the third dose booster. The immunogenicity information was from individuals’ data. There were subgroup analyses based on the anti-spike IgG status at baseline.

One limitation of our study was the small sample size for cellular immunity due to our financial limitations. We attempted to mitigate this limitation and provided a systematic way to select the sample size using a stratified random sampling technique. Another limitation was the missing data on the three-month post-vaccination follow-up. However, the missing data were less than 5% of the original cohort.

## 5. Conclusions

This study provided evidence on the promising immunogenicity of ChAdOx1 nCoV-19 as the third booster dose for the population that received two doses of CoronaVac for both humoral and cellular immunity. There was a decline in immunity level at three months post-vaccination, but the level was sufficiently high to protect against SARS-CoV-2. Subgroup analysis indicated a significantly higher neutralizing antibody improvement for population with negative anti-spike IgG at baseline.

## Figures and Tables

**Figure 1 vaccines-10-00217-f001:**
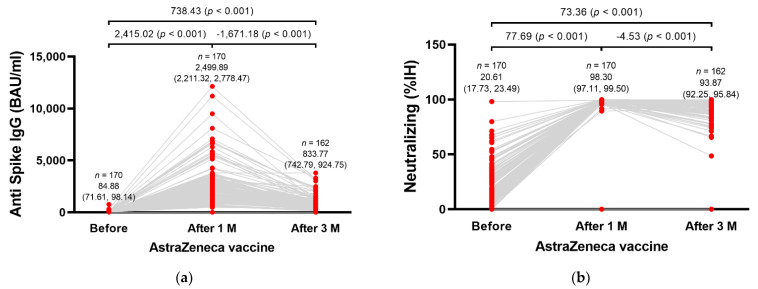
Humoral immunity profile in 170 participants after being vaccinated with ChAdOx1 nCoV-19 as the third dose following two doses of CoronaVac. (**a**) Anti-spike IgG level reported as BAU/mL and (**b**) neutralizing antibody reported as the percent of inhibition of RBD-ACE2 binding were measured before ChAdOx1 nCoV-19 vaccination (baseline), 1 month post-vaccination, and 3 months post-vaccination. BAU: binding antibody units; IgG: immunoglobulin G; IH: inhibition.

**Figure 2 vaccines-10-00217-f002:**
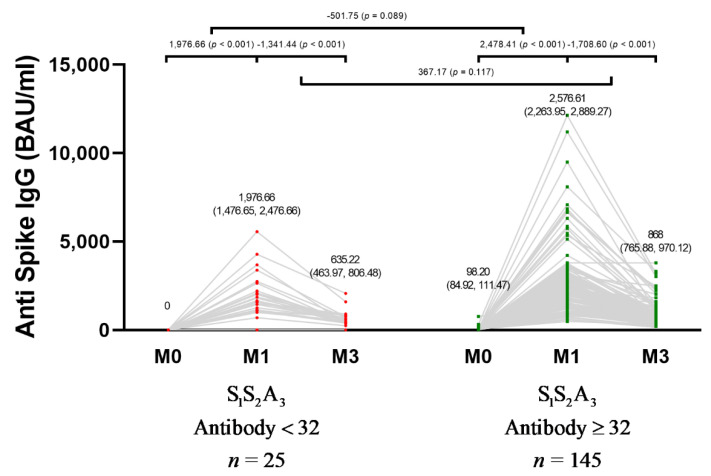
Change in anti-spike IgG level in 170 participants after received ChAdOx1 nCoV-19 as the third dose following two doses of CoronaVac based on the baseline anti-spike IgG status. Anti-spike IgG level reported as BAU/mL was measured before ChAdOx1 nCoV-19 vaccination (baseline), 1 month post-vaccination, and 3 months post-vaccination. BAU: binding antibody units; IgG: immunoglobulin G; S_1_S_2_A_3_: CoronaVac two shots followed by the third dose of ChAdOx1 nCoV-19.

**Figure 3 vaccines-10-00217-f003:**
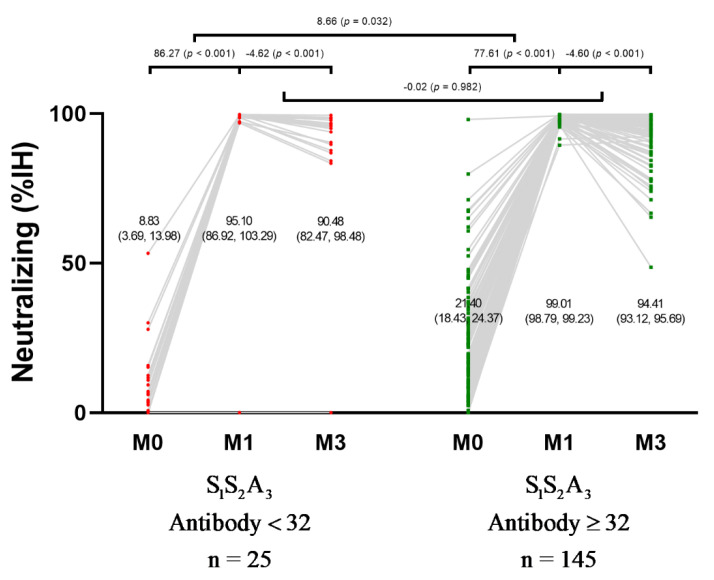
Change in neutralizing antibody level in 170 participants after received ChAdOx1 nCoV-19 as the third dose following two doses of CoronaVac based on the baseline anti-spike IgG status. Neutralizing antibody level reported as percent of inhibition of RBD-ACE2 binding was measured before ChAdOx1 nCoV-19 vaccination (baseline), 1 month post-vaccination, and 3 months post-vaccination. BAU: binding antibody units; IgG: immunoglobulin G; IH: inhibition; S_1_S_2_A_3_: CoronaVac two shots followed by the third dose of ChAdOx1 nCoV-19.

**Figure 4 vaccines-10-00217-f004:**
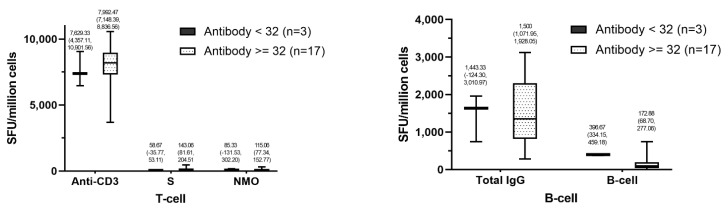
Cellular immunity level in 20 participants after being vaccinated with ChAdOx1 nCoV-19 as the third dose following two doses of CoronaVac based on the baseline anti-spike IgG status. For T cell functions, SARS-CoV-2 S defined peptide pool and SARS-CoV-2 NMO defined peptide pool were used, while ASCs counting with a sandwich assay technique was used for B cell function. T cell and B cell functions reported as SFC per one million cells were measured 3 months post-vaccination. ASCs: antibody-secreting cells; NMO: nucleoprotein, membrane protein, and open reading frame proteins; S: spike protein.

**Table 1 vaccines-10-00217-t001:** Characteristics of Participants (*n* = 170).

Characteristics	Total
Age (years)	
18–25	8 (4.71%)
26–30	18 (10.59%)
31–35	19 (11.18%)
36–40	22 (12.94%)
41–45	19 (11.18%)
46–50	33 (19.41%)
51–55	26 (15.29%)
56–60	25 (14.71%)
Female	138 (81.81%)
Smoking	8 (4.71%)
Comorbidities	72 (42.35%)
History of Vaccine Allergy	2 (1.18%)

Data were presented as counts and percentages.

## Data Availability

The supporting data for the findings of this study are available from corresponding author on reasonable request.

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
