# Peer review of "Immunogenicity of ChAdOx1 nCoV-19 Booster Vaccination Following Two CoronaVac Shots in Healthcare Workers"

_vaccines, 2022, doi:10.3390/vaccines10020217_

Round 1
Reviewer 1 Report
General remarks
In this manuscript, Wisit Prasithsirikul et al investigated the immunogenicity of healthcare workers who previously received two shots of CoronaVac and ChAdOx1 nCoV-19 as the third dose vaccine.
In the study, the authors evaluated humoral immunity of all subjects and cellular immunity of twenty subjects who were stratified random sampled based on IgG status. Both humoral and cellular immunity were improved after one-month of vaccination and subsequently declined at three months. I have no substantial requirements in terms of statistical analysis and method, but the authors need to correct errors and modify the sentences clearer.
Specific remarks
In the introduction, line 55; "within3-6 months". Please insert space between "within" and "3-6 months".
In 2.1. Study Design and Population, please explain the study period.
Line 86, "Bloom sample". please correct the spell.
Lines 89-90, "participants were separated into quartiles based on the baseline SARS-Cov-2 IgG level". Please add the specific titer of each quartile. I want to know why the number of participants with negative anti spike igG was so small in Figure 4.
Line 91, "offive". What is this word? Please choose another simple term.
In 3.1. Characteristics of Participants, lines 139-141; "The age of participants was categorized into age groups in which 46-50 years age group had the highest proportion (19.4%) and 18-25 years age group had the lowest proportion (4.7%)." This sentence is a little complicated. Please modify it simpler and clearer.
In 3.1. Characteristics of Participants, please add the median or mean age of the subjects.
Line 143, "73 (42.9%) participants who had comorbidities." Please explain the comorbidities in detail (such as hypertension, diabetes, obesity, autoimmune disease,....).
In the Characteristics of Participants, it is better to add the participants' past medical history of Covid-19. Is it available?
In figure 1, all subjects received AstraZeneca vaccine. In figure 2/3, all subjects received S1S2A3. Are these annotations necessary?
Lines 183-184 "insufficient evidence that the improvement was significantly different between participants with positive and negative baseline anti spike IgG (p=0.089)." Is "the improvement was not significantly different between participants with positive and negative baseline anti spike IgG (p=0.089)." better?
Lines 184-188 "The anti spike IgG at 3 months postvaccination was significantly decreased to 635.22 BAU/ml (95% CI, 463.97-806.48) when compared to IgG level at 1 month postvaccination in negative baseline IgG group (p<0.001), " This sentence is a little complicated. Please modify it clearer.
In Figure 4, the range of box plot is narrow so it is difficult to distinguish two groups by the design pattern of the box. Please modify the figure clearer (e.g. omit "Positive" and change the scale.)
In Figure 4, If the authors would not omit "Positive" ,please explain what "Positive" means.
Lines 255-256 "a higher B cell response in participants with negative IgG at baseline were detected." The correct sentence is "a higher B cell response in participants with negative IgG at baseline was detected."
Lines 283-287, "On the other hand, there was a randomized controlled trial that reported poor improvement of cellular immunity in homologous three doses of ChAdOx1 nCoV-19 while indicated good improvement of cellular immunity at one-month postvaccination in heterologous ChAdOx1 nCoV-19 as third dose booster in participants who received two shots of BNT162b2 which had a similar result to the heterologous ChAdOx1 nCoV-19 booster after fully vaccinated with CoronaVac in this study." This sentence is very complicated and difficult. Please modify it simpler.
Author Response
Reviewer 1: In this manuscript, Wisit Prasithsirikul et al investigated the immunogenicity of healthcare workers who previously received two shots of CoronaVac and ChAdOx1 nCoV-19 as the third dose vaccine. In the study, the authors evaluated humoral immunity of all subjects and cellular immunity of twenty subjects who were stratified random sampled based on IgG status. Both humoral and cellular immunity were improved after one-month of vaccination and subsequently declined at three months. I have no substantial requirements in terms of statistical analysis and method, but the authors need to correct errors and modify the sentences clearer.
Response: Thank you very much for your comments and suggestions.
Reviewer 1: In the introduction, line 55; "within3-6 months". Please insert space between "within" and "3-6 months".
Response: Corrected.
Reviewer 1: In 2.1. Study Design and Population, please explain the study period.
Response: The study period was added.
Reviewer 1: Line 86, "Bloom sample". please correct the spell.
Response: Corrected.
Reviewer 1: Lines 89-90, "participants were separated into quartiles based on the baseline SARS-Cov-2 IgG level". Please add the specific titer of each quartile. I want to know why the number of participants with negative anti spike igG was so small in Figure 4.
Response: The titer of each quartile was added.
Reviewer 1: Line 91, "offive". What is this word? Please choose another simple term.
Response: Corrected.
Reviewer 1: In 3.1. Characteristics of Participants, lines 139-141; "The age of participants was categorized into age groups in which 46-50 years age group had the highest proportion (19.4%) and 18-25 years age group had the lowest proportion (4.7%)." This sentence is a little complicated. Please modify it simpler and clearer.
Response: The sentence was revised.
Reviewer 1: In 3.1. Characteristics of Participants, please add the median or mean age of the subjects.
Response: The mean, median, and IQR were added.
Reviewer 1: Line 143, "73 (42.9%) participants who had comorbidities." Please explain the comorbidities in detail (such as hypertension, diabetes, obesity, autoimmune disease,....).
Response: The co-morbidities were elaborated.
Reviewer 1: In the Characteristics of Participants, it is better to add the participants' past medical history of Covid-19. Is it available?
Response: None of the participants has a history of Covid-19 infection. The statement was added.
Reviewer 1: In figure 1, all subjects received AstraZeneca vaccine. In figure 2/3, all subjects received S1S2A3. Are these annotations necessary?
Response: We believe the annotations are useful and would like to keep it as is.
Reviewer 1: Lines 183-184 "insufficient evidence that the improvement was significantly different between participants with positive and negative baseline anti spike IgG (p=0.089)." Is "the improvement was not significantly different between participants with positive and negative baseline anti spike IgG (p=0.089)." better?
Response: Corrected as advised.
Reviewer 1: Lines 184-188 "The anti spike IgG at 3 months postvaccination was significantly decreased to 635.22 BAU/ml (95% CI, 463.97-806.48) when compared to IgG level at 1 month postvaccination in negative baseline IgG group (p<0.001), " This sentence is a little complicated. Please modify it clearer.
Response: The sentence was revised.
Reviewer 1: In Figure 4, the range of box plot is narrow so it is difficult to distinguish two groups by the design pattern of the box. Please modify the figure clearer (e.g. omit "Positive" and change the scale.)
Response: The figure was revised.
Reviewer 1: In Figure 4, If the authors would not omit "Positive" ,please explain what "Positive" means.
Response: The “Positive” terms in Figure 4 were changed to Anti-CD3 for T cells and Total IgG for B cells. The format of Figure 4 was revised.
Reviewer 1: Lines 255-256 "a higher B cell response in participants with negative IgG at baseline were detected." The correct sentence is "a higher B cell response in participants with negative IgG at baseline was detected."
Response: Corrected.
Reviewer 1: Lines 283-287, "On the other hand, there was a randomized controlled trial that reported poor improvement of cellular immunity in homologous three doses of ChAdOx1 nCoV-19 while indicated good improvement of cellular immunity at one-month postvaccination in heterologous ChAdOx1 nCoV-19 as third dose booster in participants who received two shots of BNT162b2 which had a similar result to the heterologous ChAdOx1 nCoV-19 booster after fully vaccinated with CoronaVac in this study." This sentence is very complicated and difficult. Please modify it simpler.
Response: The sentence was revised.
Reviewer 2 Report
This paper investigates immunogenicity of the Astra-Zeneca booster following two vaccinations with the Sinovac CoronaVac vaccine in healthcare workers in Thailand. This study comes at a time when booster vaccinations take place all over the world and represents an important contribution to understanding the immunogenicity of nonhomologous combinations of anti-COVID19 vaccines.
While I find the study as a whole sound and significant, I have a problem with Figure 4 (cellular immunity).
- The y-axis label “Units” is insufficient – which units? (SFC per 1 million cells?)
- What are “Positive” in this figure? This should be explained in the text and their results commented upon. Their T cell and B cell results are much higher than for the rest and should probably be on a separate graph with a different scale.
- The boxes for Ab < 32 and Ab >=32 results should be chosen in different colours (both outline and fill) instead of black outline and grayscale fill, so that they can be distinguished.
It would be helpful to have a more detailed comparison with 3 x Sinovac and 2 x Sinovac with Pfizer booster (with concrete numbers from the literature).
Author Response
Reviewer 2: This paper investigates immunogenicity of the Astra-Zeneca booster following two vaccinations with the Sinovac CoronaVac vaccine in healthcare workers in Thailand. This study comes at a time when booster vaccinations take place all over the world and represents an important contribution to understanding the immunogenicity of nonhomologous combinations of anti-COVID19 vaccines.
Response: Thank you very much for your comments and suggestions.
Reviewer 2: While I find the study as a whole sound and significant, I have a problem with Figure 4 (cellular immunity). The y-axis label “Units” is insufficient – which units? (SFC per 1 million cells?) What are “Positive” in this figure? This should be explained in the text and their results commented upon. Their T cell and B cell results are much higher than for the rest and should probably be on a separate graph with a different scale. The boxes for Ab < 32 and Ab >=32 results should be chosen in different colours (both outline and fill) instead of black outline and grayscale fill, so that they can be distinguished.
Response: Figure 4 was revised.
Reviewer 2: It would be helpful to have a more detailed comparison with 3 x Sinovac and 2 x Sinovac with Pfizer booster (with concrete numbers from the literature).
Response: Pfizer vaccine was not readily available at the inception of this study so we did not have the comparative data between the two vaccination profiles.